## Original Research Article

*Arabidopsis thaliana*; *image quantification*; *live-cell imaging*; *mitochondria*; *zygote*.

**Author for correspondence:**
M. Ueda, Tel.: +81 22-795-6713;
E-mail: minako.ueda.e7@tohoku.ac.jp

# Mitochondrial dynamics and segregation during the asymmetric division of *Arabidopsis* zygotes

Yusuke Kimata[1,2], Takumi Higaki[3], Daisuke Kurihara[1,4], Naoe Ando[5], Hikari Matsumoto[5], Tetsuya Higashiyama[1,5,6], and Minako Ueda[1,2,5]

[1]Institute of Transformative Bio-Molecules (ITbM), Nagoya University, Furo-cho, Chikusa-ku, Nagoya, Aichi 464-8601, Japan; [2]Graduate School of Life Sciences, Tohoku University, Sendai, 980-8578, Japan; [3]International Research Organization for Advanced Science and Technology (IROAST), Kumamoto University, 2-39-1 Kurokami, Chuo-ku, Kumamoto 860-8555, Japan; [4]JST, PRESTO, Furo-cho, Chikusa-ku, Nagoya, Aichi 464-8601, Japan; [5]Division of Biological Science, Graduate School of Science, Nagoya University, Furo-cho, Chikusa-ku, Nagoya, Aichi 464-8602, Japan; [6]Department of Biological Sciences, Graduate School of Science, University of Tokyo, 7-3-1 Hongo, Bunkyo-ku, Tokyo 113-0033, Japan

## Abstract

The zygote is the first cell of a multicellular organism. In most angiosperms, the zygote divides asymmetrically to produce an embryo-precursor apical cell and a supporting basal cell. Zygotic division should properly segregate symbiotic organelles, because they cannot be synthesized *de novo*. In this study, we revealed the real-time dynamics of the principle source of ATP biogenesis, mitochondria, in *Arabidopsis thaliana* zygotes using live-cell observations and image quantifications. In the zygote, the mitochondria formed the extended structure associated with the longitudinal array of actin filaments (F-actins) and were polarly distributed along the apical–basal axis. The mitochondria were then temporally fragmented during zygotic division, and the resulting apical cells inherited mitochondria at higher concentration compared to the basal cells. Further observation of postembryonic organs showed that these mitochondrial behaviours are characteristic of the zygote. Overall, our results showed that the zygote has spatiotemporal regulation that unequally distributes the mitochondria.

## 1. Introduction

In most plant species, the multicellular body is generated from a unicellular zygote. In *Arabidopsis thaliana*, the apical–basal (shoot–root) axis is already evident at the asymmetric division of the zygote, which generates one small apical cell and one large basal cell (Mansfield & Briarty, 1991). The apical cell actively proliferates to produce most of the embryonic organs, whereas the basal cell mostly yields the extra-embryonic suspensor (Gooh et al., 2015). This early embryogenesis would require massive energy production, because the principle source of ATP biogenesis, mitochondria, are necessary at this stage. In *Arabidopsis thaliana*, a mutation in the mitochondria-localizing protein, MIRO1 GTPase, causes growth arrest just after zygotic division (Yamaoka et al., 2011). Various other mitochondria-related mutants also arrest growth within several rounds of zygotic divisions, confirming that mitochondria are important for the initiation phase of plant ontogeny (Deng et al., 2014; He et al., 2017; Yamaoka & Leaver, 2008).

Furthermore, the apical cell lineage has higher proliferation activity than the basal cell and initiates various developmental programmes to generate diverse embryonic organs, implying a higher energy demand (Gooh et al., 2015). Because mitochondria cannot be synthesized *de novo*, it would be important to provide the apical cell at the zygotic division with sufficient starting mitochondria. Indeed, Gao et al. (2018) showed that the apical cell increases the mitochondrial number already before its cell division. However, the mechanism by which the plant zygote distributes the mitochondria into the two daughter cells is still unknown, as is whether the mitochondria are inherited unequally. Recently, we used live-cell imaging of *Arabidopsis* zygotes and found that the vacuoles are polarly distributed in the zygote, causing their unequal inheritance into the basal daughter cell (Kimata et al., 2019). In the zygote, the actin filaments (F-actin) align longitudinally along the apical–basal axis and the thin vacuolar strands associate with the F-actin cables to accumulate at the basal cell end (Kimata et al., 2016; 2019).

In this study, we utilized live-imaging system to visualize the mitochondrial dynamics in the *Arabidopsis* zygote. We found that the mitochondrial distribution pattern in the egg cell is lost upon fertilization, after which time the mitochondria associate with the F-actin to form a filamentous network. After polar distribution along the apical–basal axis is completed, the filamentous mitochondria were temporally fragmented during zygotic division, then segregated into the apical and basal daughter cells at different densities. Similar features were

not observed in other tissues, such as leaf trichomes and root meristems, although the densely packed mitochondria were detected after the asymmetric cell division of the stomatal precursor cell. Overall, our findings show that the *Arabidopsis* zygote polarly distributes the mitochondria along the apical–basal axis, and concentrates the mitochondria into the embryo's initial apical cell.

## 2. Methods

### 2.1. Strains and growth conditions

All *Arabidopsis* lines were in the Columbia (Col-0) background. Plants were grown at 18–22°C under continuous light or long-day conditions (16-h light/8-h dark). The mitochondrial/nuclear marker for egg cell and zygote contains DD45p::mt-GFP (Yamaoka et al., 2011) as a mitochondria-localizing reporter, and EC1p::H2B-tdTomato and DD22p::H2B-mCherry as zygote and endosperm nuclear reporters, respectively (coded as MU1968) (Kimata et al., 2019). DD45p::mt-GFP consists of DD45 promoter and a mitochondrial targeting (mt) signal peptide, which is fused to green-fluorescent protein (GFP). EC1p::H2B-tdTomato includes EGG CELL1 (EC1) promoter (Sprunck et al., 2012), histone 2B (H2B), and red-fluorescent tandem dimer Tomato (tdTomato). DD22p::H2B-mCherry contains DD22 promoter, H2B, and red-fluorescent mCherry.

### 2.2. Plasmid construction

The F-actin/mitochondrial marker contains EC1p::Lifeact-mTur2 (coded as MU1984) and DD45p::mt-Kaede (Hamamura et al., 2011). In EC1p::Lifeact-mTur2, the 463-bp EC1 promoter was fused to Lifeact sequence from the previously reported actin marker EC1p::Lifeact-Venus (Kawashima et al., 2014), cyan-fluorescent mTurquoise2 (mTur2), and NOPALINE SYNTHASE (NOS) terminator in a pMDC99 binary vector (Curtis & Grossniklaus, 2003).

The mitochondrial/DRP3A marker contains DD45p::mt-tdTomato (coded as MU2391) and EC1p::DRP3A-Clover (MU2407). In DD45p::mt-tdTomato, the 2,409-bp DD45 promoter was fused to mt signal peptide from DD45p::mt-GFP, tdTomato, and NOS terminator in a pMDC100 binary vector (Curtis & Grossniklaus, 2003), in which Kanamycin-resistant *NPTII* gene was replaced by Gentamycin-resistant *aacC1* gene of pPZP221 binary vector (Hajdukiewicz et al., 1994). In EC1p::DRP3A-Clover, EC1 promoter was fused to the full-length coding region of DRP3A, green-fluorescent Clover, and NOS terminator in a pMDC99 binary vector.

The mitochondrial/nuclear marker for postembryonic organs contains RPS5Ap::mt-GFP (coded as MU2184) and RPS5Ap::H2B-tdTomato (Kimata et al., 2019). In RPS5Ap::mt-GFP, 1.7-kb RIBOSOMAL PROTEIN SUBUNIT 5A (RPS5A) promoter (Weijers et al., 2001) was fused to mt signal peptide, GFP, and NOS terminator in a pPZP221 binary vector (Hajdukiewicz et al., 1994).

These constructs were transformed into *Arabidopsis* using floral dip method (Clough & Bent, 1998).

### 2.3. Time-lapse observation and histological analysis

*In vitro* ovule culturing and live-cell imaging of the zygote were performed as previously described (Gooh et al., 2015; Kimata et al., 2016; Kurihara et al., 2017; Ueda et al., 2020). For inhibitor treatment of latrunculin B (LatB) and oryzalin, 1 $\mu$M reagent dissolved in 0.1% dimethyl sulfoxide (DMSO) was added to the ovule cultivation media. For the observation of postembryonic organs, seedlings were stained with 10 $\mu$g/ml propidium iodide (PI) solution.

The two-photon excitation microscopy (2PEM) images were all acquired using a laser-scanning inverted microscope (A1R MP; Nikon) equipped with a Ti:sapphire femtosecond pulse laser (Mai Tai DeepSee; Spectra-Physics) and a 40× water-immersion objective lens (CFI Apo LWD WI, NA = 1.15, WD = 0.59–0.61 mm; Nikon) with Immersol W 2010 (Zeiss). All fluorescent signals were detected at 950 nm excitation, only except for the F-actin/mitochondrial marker, whose signal was acquired at 920 nm. We used two dichroic mirrors, DM495 and DM560, and three band-pass filters: 479/40 nm for mTur2, 534/30 nm for GFP, Clover and Kaede, and 578/105 nm for tdTomato, mCherry, and PI. Time-lapse images were taken in 512 × 512 pixels with 3× zoom and 1-$\mu$m z-intervals every 10 or 20 min, except for the quantitative analysis, which was done with the time-lapse images taken in 1,024 × 1,024 pixels with 4× zoom and 0.25-$\mu$m z-intervals every 20 min. High-resolution snapshots for the quantitative analysis of DMSO- and LatB-treated zygotes and postembryonic organs were taken in 1,024×1,024 pixels with 8× zoom and 1-$\mu$m z-intervals, except for stomatal precursor cells, which were observed with 10× zoom. These snapshots and the images of mitochondrial/DRP3A marker were shown after processing with "2D Deconvolution" function in NIS-Elements AR 5.21 software (Nikon). For the measurement of mitochondrial association with F-actin, the snapshots of F-actin/mitochondrial marker were processed with "Denoise" and "2D Deconvolution" functions in NIS-Elements AR 5.21 software, and then used for quantification (see below section). Maximum intensity projection (MIP) was performed using ImageJ (http://rsbweb.nih.gov/ij/index.html).

### 2.4. Image analyses and statistical tests

For quantification of mitochondrial area, a 1–4 pixel-band-pass filter was applied to the mt-GFP images to perform noise reduction and mitochondrial shape enhancement. LPX ImageJ plugins (https://lpixel.net/services/research/lpixel-imagej-plugin/) were used for band-pass filtering. Then, the filtered images were binarized by the Otsu's algorithm. The cell areas were manually determined based on the cell outline (Figure 1e), and the binary images were masked with the cell area (Figure 1f). Mitochondrial occupancy was defined and calculated as the ratio of the total mitochondrial area to the cell area.

To measure the average angle of zygotic mitochondria against the cell longitudinal axis ($\Delta\theta$), the mt-GFP images and the cell medial axis images (Akita et al., 2015) in the cell center regions, which were defined as a circle with a radius of 200 pixels, which corresponds to 15.6 $\mu$m, centered on the midpoint of the cell medial axis (Figure 2a–c) were applied to the ImageJ plugin Directionality (https://imagej.net/Directionality). $\Delta\theta$ was defined as

$$\begin{cases} |\theta_{\text{cell}} - \theta_{\text{mito}}| & \text{if } |\theta_{\text{cell}} - \theta_{\text{mito}}| \leq 90 \\ |\theta_{\text{cell}} - \theta_{\text{mito}}| - 90 & \text{if } |\theta_{\text{cell}} - \theta_{\text{mito}}| > 90 \end{cases},$$

where $\theta_{\text{mito}}$ and $\theta_{\text{cell}}$ are the mitochondrial angle and the cell longitudinal angle.

To quantify the mitochondrial association with F-actin, the intensity of F-actin signals in mitochondrial regions was measured using deconvoluted F-actin/mitochondrial marker images

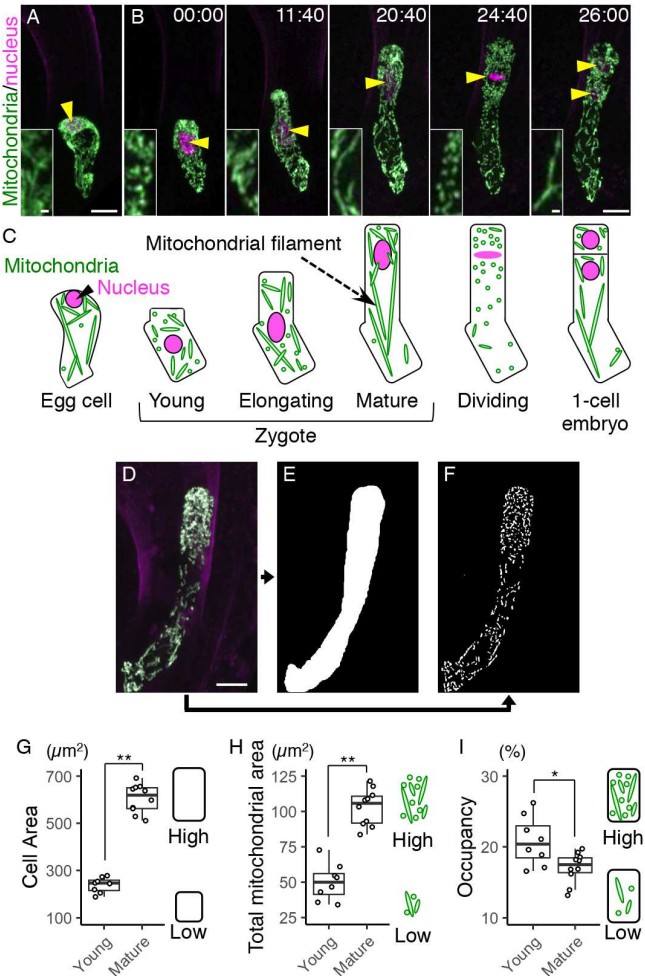

**Fig. 1.** Live-cell imaging and quantification of zygotic mitochondria. (a and b) Two-photon excitation microscopy (2PEM) images of the egg cell (a) and the time-lapse observation of the zygote in the *in vitro*-cultivated ovules (b) expressing the mitochondrial/nuclear marker. Maximum intensity projection (MIP) images are shown. Images are representative of nine time-lapse observations. Numbers indicate the time (h:min) from when the zygote started elongation. Yellow arrowheads indicate the nuclei, and the inset shows an enlarged image of the basal cell region. (c) Illustrations showing a summary of the respective stages. (d–f) Image processing for quantification analysis. (d) MIP image generated by serial optical sections of 2PEM images of a mature zygote expressing the mitochondrial/nuclear marker. (e) Mask image of the cell area. (f) Binary image of the mitochondria. (g–i) Graphs of the cell area (g), the total mitochondrial area (h) and the mitochondrial occupancy (i) in the young and mature zygotes. Right illustrations show the correlation between the values and the features of cells and/or mitochondria. Significant difference was determined by Mann–Whitney $U$ test; $^*p < .05$; $^{**}p < .01$ [$n = 8$ (young) and 10 (mature)]. Scale bars: 10 and 1 $\mu$m (insets).

(see above). For segmentation of mitochondrial regions, a band-pass filter (2–6 pixels), binarization by the Yen's algorithm, and watershed separation were applied to mt-Kaede images. For shape enhancement of F-actin, the background signal in each Lifeact-mTur2 image was reduced using "Subtract Background" function (Rolling ball radius = 50 pixels). Finally, the association ratio was defined as the proportion of mitochondria overlapping with F-actin signals over 500 [a. u.], which corresponded to the mean intensity of fine F-actin fibers.

For measurements of Feret's diameter and circularity ($4\pi \times$Area/Perimeter length$^2$) of mitochondria before, during, and after the zygotic division, the "analyze particles" function in ImageJ was used to the above images. To measure the mitochondrial circularity and

Feret's diameter in DMSO- and LatB-treated zygotes and postembryonic cells except for stomatal precursor cell, band-pass filters (6–12 pixels for the zygote and root hair; 8–12 pixels for the epidermal cell and trichome; 5–20 pixels for the root meristematic cell) were applied to their mitochondrial images of single focal plane, and the filtered images were binarized by the Otsu's algorithm. Watershed separation was also applied to these binarized images. Then, the mean circularity and Feret's diameter of mitochondria in each cell were calculated.

To measure the shape and occupancy of mitochondria in the stomatal meristemoid and stomatal lineage ground cell (SLGC), 6–20 pixel-band-pass filter was applied to MIP images. The filtered images were binarized by the Otsu's algorithm, and watershed separation was also applied to the binarized images. Finally, the mean circularity and Feret's diameter of mitochondria in each cell were measured, and the mitochondrial occupancy was calculated by dividing the mitochondrial area by manually determined cellular area.

To examine distribution of the mt-GFP intensity in the zygotes and one-cell embryos, the major axis of the fitted ellipse of the cell was defined as apical–basal axis. Then, we obtained the profile of the average GFP intensity perpendicular to the axis in the cell using LPX ImageJ plugins.

All image processing and analyses were performed using ImageJ (http://rsbweb.nih.gov/ij/index.html). Statistical analyses were performed using R (ver. 3.6.1: https://www.r-project.org/). "multcomp" package (http://multcomp.r-forge.r-project.org/) was used for Tukey–Kramer and Dunnett's test.

## 3. Results

### 3.1. The mitochondrial pattern is randomized at fertilization and mitochondrial area gradually increases during zygote elongation

To perform live-cell imaging of the mitochondria in *Arabidopsis* zygotes, we combined the green-fluorescent reporter visualizing mitochondria (DD45p::mt-GFP) and the red-fluorescent reporter labelling nucleus (EC1p::H2B-tdTomato and DD22p::H2B-mCherry) (Kimata et al., 2019; Yamaoka et al., 2011). This dual-colour mitochondrial/nuclear marker was imaged using 2PEM (Figure 1a, b, and Supplementary Movie S1).

Before fertilization, the mitochondria were in a mesh-like pattern outside of the cellular centre (Figure 1a, c), probably because this region was occupied by large vacuoles (Kimata et al., 2019; Mansfield et al., 1991). After fertilization, the mesh-like pattern was lost, and the mitochondria were detected in the cell centre (Figure 1b, c), as the large vacuoles had shrunk (Faure et al., 1992; Kimata et al., 2019). Then the cell elongated along the apical–basal axis, and the total mitochondrial area seemed to gradually increase (Figure 1b,c, and Supplementary Movie S1).

We then began quantifying the changes in zygotic and mitochondrial areas. To distinguish between individual mitochondria, we acquired images with a higher spatial resolution than that used for Figure 1a, b (seven times the *xy* resolution and four times the *z* resolution), and used the images (Figure 1d) to extract the cell area (Figure 1e) and the mitochondria (Figure 1f). The cell and total mitochondrial areas were both higher in mature zygotes than in young zygotes (Figure 1g, h). The mitochondrial occupancy was slightly decreased in mature zygotes (Figure 1i), which is consistent with our previous finding that the zygote quickly expands with massive vacuole swelling (Kimata et al., 2019).

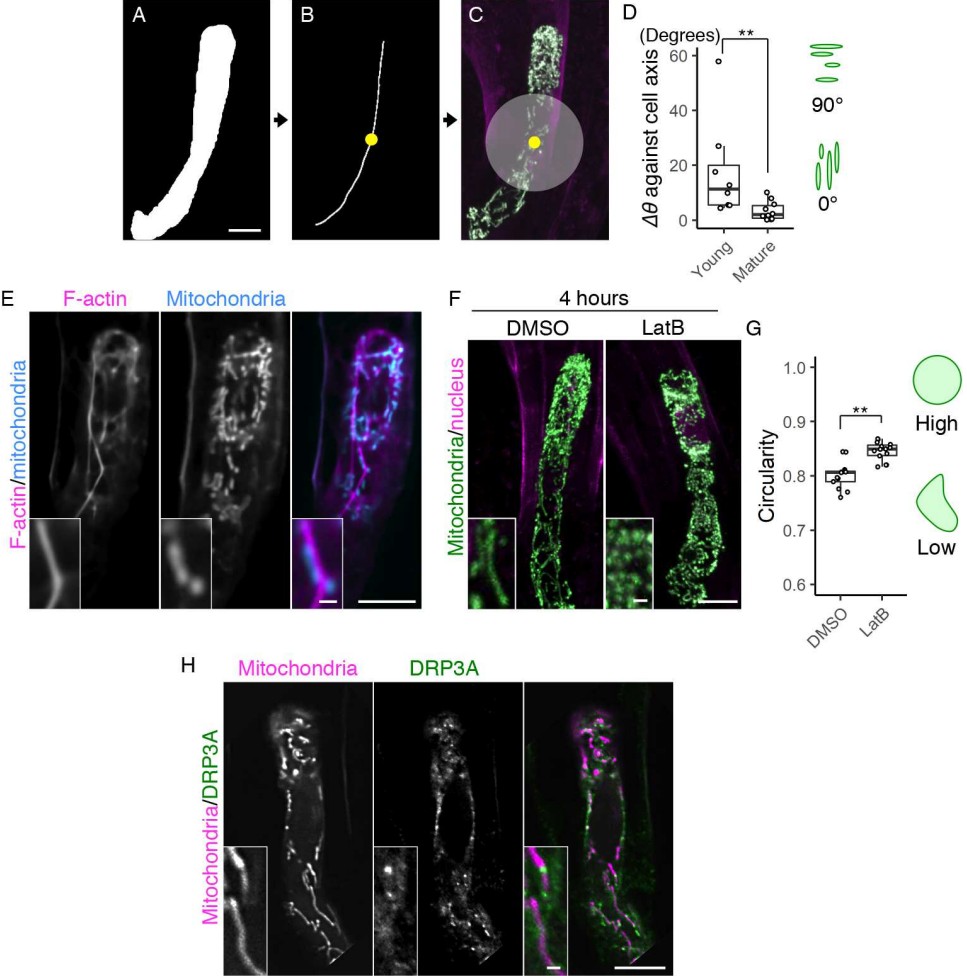

**Fig. 2.** Formation of the filamentous mitochondria and its dependence on F-actin cables. (a–c) Image processing for quantification analysis. (a) Mask image of the cell area, shown as in Figure 1e. (b) Cell medial axis image. The yellow dot shows the midpoint of the axis. (c) The centre region, defined as a circle with a radius of 15.6 $\mu$m (200 pixels), merged with Figure 1d. (d) Graph of $\Delta\theta$ (the average angle of the mitochondria) against the cell longitudinal axis, which was measured in centre regions (c) of the mature and young zygotes. Note that the entire cell area in all tested young zygotes was included in the circle. (e) Two-photon excitation microscopy (2PEM) images of the mature zygote expressing the F-actin/mitochondrial marker. Maximum intensity projection (MIP) images are shown, and insets show the enlarged images of the basal cell region. (f) 2PEM images of the mitochondrial/nuclear marker after the exposure to the control dimethyl sulfoxide (DMSO) and the polymerization inhibitor of F-actin (1 $\mu$M LatB) for 4 h. (g) Graph of the circularity, which was measured using the mitochondria on the single focal plane of entire cell regions of the DMSO- and LatB-treated zygotes. The right illustrations show the correlation between the mitochondrial shape and value. (h) 2PEM images of the mitochondrial/DRP3A marker. The midplane images are shown, and insets show enlarged images of the basal cell region. Significant differences were determined by Mann–Whitney $U$ test (d) and Student's $t$ test (g); $^{**}p < .01$ [$n = 8$ (young) and 10 (mature) in d, and $n = 13$ (DMSO) and 17 (LatB) in g]. Scale bars: 10 and 1 $\mu$m (insets).

## 3.2. Zygotic mitochondria form a long filamentous structure along the F-actin array

During the live-cell observation, we noticed that mitochondria in elongating zygotes formed long filamentous structures along the apical–basal axis (Figure 1b, c, and Supplementary Movie S1). We therefore measured the average angle of the mitochondria against the cell longitudinal axis ($\Delta\theta$) (Figure 2a–d). The basal bottom of the zygote was often bent (Figure 1d), so we excluded this site and focussed on the cell's centre region (Figure 2c, grey circle). The mitochondrial angle in the mature zygote was significantly smaller than in the young zygote, showing the longitudinal direction of mitochondrial filaments (Figure 2d).

To identify the driving force behind this mitochondrial alignment, we assessed the involvement of F-actin, because we previously found a similar longitudinal array of F-actin in the zygote (Kimata et al., 2016). We constructed a dual-colour marker that simultaneously visualized F-actin and mitochondria (F-actin/mitochondrial marker), and found that they were associated with each other in mature zygotes (Figure 2e). The ratio of mitochondria associated with F-actin cables was 59.5 ± 14.3% ($n = 3$). We further tested the effect of the actin polymerization inhibitor latrunculin B (LatB), which effectively destroyed the F-actin array in our previous zygote observation (Kimata et al., 2016). After treatment with LatB, the mitochondrial filaments disappeared, leaving clusters of dotted mitochondria (Figure 2f) that had higher mitochondrial circularity values (Figure 2g). Therefore, we concluded that zygotic mitochondria form longitudinal filaments in an F-actin-dependent manner.

In both animals and plants, the mitochondrial morphology is usually determined by a balance between mitochondrial fission and fusion (Arimura, 2018). To assess the involvement of the fission/fusion regulation in the zygote, we focussed on a crucial fission regulator, DYNAMIN-RELATED PROTEIN 3A (DRP3A), because no fusion regulator has been identified in plants (Arimura

et al., 2004; Arimura, 2018; Fujimoto et al., 2009). We found high levels of DRP3A expression in the published RNA-seq database of the *Arabidopsis* zygote (Zhao et al., 2019). In this database, the FPKM (fragments per kilobase of transcript, per million mapped reads) values of expressed genes are estimated to be >1. The value of DRP3A (60.1) is comparable to those of ZYGOTE-ARREST1 (53.0), which is known as ANAPHASE-PROMOTING COMPLEX/CYCLOSOME 11 (APC11), and thus is essential for zygotic division (Guo et al., 2016), and WRKY2 transcription factor (58.0), which induces zygotic transcription to polarize the zygote (Ueda et al., 2017), implying that fission/fusion machinery may contribute to zygote development. Therefore, we generated a dual-colour mitochondrial/DRP3A marker, which contains DD45p::mt-tdTomato and EC1p::DRP3A-Clover, to examine the co-localization of this machinery (Figure 2h). We found bright DRP3A dots between the mitochondria, consistent to the finding that DRP3A localizes at the mitochondrial fission sites in leaf epidermal cells (Fujimoto et al., 2009). These bright dots were not, however, detected on the filamentous mitochondria, implying that fission/fusion activity was biased in forming long connected mitochondria during zygote elongation.

### 3.3. Filamentous mitochondria are temporally fragmented during the zygotic division

In the time-lapse movies of the zygote, we also noticed that many dotted mitochondria were observed instead of the longitudinal filaments during cell division (Figure 1b, c, and Supplementary Movie S1); the filamentous structures were detected again after zygotic division was completed. To further focus on this temporal mitochondrial fragmentation, we increased the time resolution. We initially performed conventional time-lapse imaging with long time intervals (20 min) to observe the period from the emergence of the young zygote until cell division without causing severe laser damage (Supplementary Movie S1) (Kurihara et al., 2017), but with this interval, cell division was captured only in 1 or 2 frames; we thus divided the interval in half (10 min), focussing just on the periods before and after the zygotic division (Supplementary Movie S2). As a result, we found that the formation of spherical mitochondria was tightly synchronized with the cell division period (Figure 3a and Supplementary Movie S2).

To quantify this correlation, we evaluated the changes in mitochondrial sphericity by measuring the max/(max + min) ratio of Feret's diameter (Figure 3b) and the circularity (Figure 3c) in the cell centre regions (circled area in Figure 2c), focussing on before, during, and after zygotic division (i.e., mature, dividing, and one-cell embryo, respectively). Only the dividing zygotes showed a reduced max/(max + min) ratio of Feret's diameter and increased circularity, confirming that this is the phase of zygotic division at which spherical mitochondria form (Figure 3b,c). In contrast, the total mitochondrial area in the whole cell was similar at all stages tested (Figure 3d), implying the involvement of mitochondrial fission, which does not affect the total volume (Arimura, 2018). This is also consistent with our previous time-lapse observation, which showed that the F-actin array itself is not altered during zygotic division (Kimata et al., 2016).

We also tested whether the mitochondrial fragmentation depends on the cell division phase, or just occurs after zygote elongation. Therefore, we applied the microtubule polymerization inhibitor oryzalin, which blocks proper spindle formation and thus prolongs the cell division period in our imaging system (Kimata et al., 2016). The mitochondria kept their dotted shape during this period (Figure 3e and Supplementary Movie S3), showing that the mitochondrial fragmentation is linked to the cell division phase, and not just induced after zygote elongation.

### 3.4. Mitochondria are polarly distributed in the zygote and differently segregate into the apical and basal daughter cells

Our findings on the long filamentous mitochondrial formation and its temporal separation during cell division raised the idea that the zygote might polarly position the mitochondria along the apical–basal axis, then unequally segregate them into the two daughter cells. To test this idea, we examined mitochondrial distribution along the apical–basal axis in young zygotes (Figure 4a), and the zygotes before and after cell division (mature and one-cell embryo; Figure 4b,c). In young zygotes, the mitochondria symmetrically distributed and slightly reduced in the cellular centre, probably because the nucleus positioned in this region (Figure 4a). In contrast, the mitochondria accumulated in the apical region in mature zygotes (Figure 4b), showing a polar distribution along the apical–basal axis. Similar distribution pattern was observed after zygotic division (Figure 4c), suggesting that the polar mitochondrial positioning in the zygote pre-sets the unequal segregation of the mitochondria into the apical and basal daughter cells. Taken together, we concluded that the zygote spatiotemporally regulates the inheritance of densely packed mitochondria into the apical cells, which are the initial cells of plant ontogeny (Figure 4d).

### 3.5. Filamentous mitochondrial formation during cell elongation and temporal fragmentation during cell division are characteristic of the zygote

We then examined whether the morphological features that we found in the zygote are specific or common to other tissues. As the typical tissues showing remarkable cell elongation like in the zygote, we focussed on the root hair, root epidermis, and leaf trichome (Figure 5a–h). In addition, we also tested the root meristem which, like the embryo, is a highly proliferating tissue (Figure 5i). In the elongated root hair, the small dotted mitochondria were localized under the cell surface (Figure 5a,b). This pattern agrees with a previous observation (Zheng et al., 2009), and large vacuoles in the cell centre would explain the exclusion of the mitochondria from this area (Kimata et al., 2019). In the differentiated root epidermis, the mitochondria were larger than those in the root hair, and also excluded from the cell centre (Figure 5c–e), whereas the mitochondria in the leaf trichome were detected in the central region (Figure 5f–h). In spite of their different sizes and distributions, the predominant mitochondrial shape was spherical in all cell types, as confirmed by their higher circularity values and lower ratio of max/(max + min) Feret's diameter than those of the zygote (Figure 5j,k). In the root meristems, the mitochondria were not spherical, but more compact than in the zygote, as shown by the similar circularity value and lower ratio of Feret's diameter than in the zygote (Figure 5i,j,k). In the root meristem, we also found that mitochondrial fragmentation does not occur during cell division (Figure 5i), and this was confirmed by the similar mitochondrial values in dividing and non-dividing cells (Figure 5l,m).

We also observed the stomatal precursor cells, which divide asymmetrically to generate a small stem cell-like meristemoid and a large SLGC (Figure 5n). We found that the mitochondrial occupancy was higher in the meristemoid than in the SLGC (Figure 5o). In contrast to the basal cell after zygotic division, the SLGC did not contain filamentous mitochondria (Figure 5n, compare to

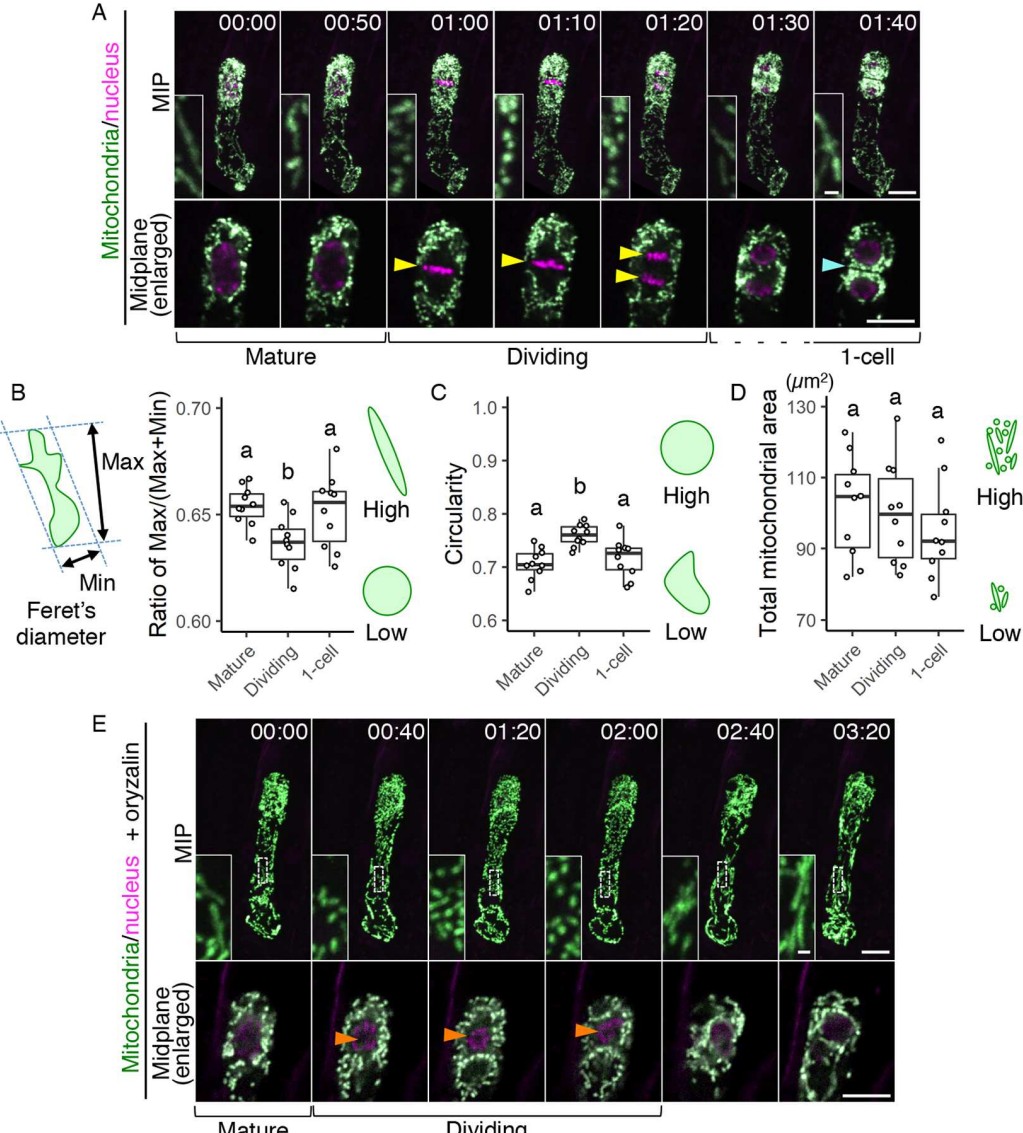

**Fig. 3.** Temporary mitochondrial fragmentation during zygotic division. (a) Two-photon excitation microscopy (2PEM) images of the time-lapse observation of the mitochondrial/nuclear marker. Images are representative of eight time-lapse observations. Numbers indicate the time (h:min) from when the observation began, and the inset shows an enlarged image of the basal region. Upper panels show maximum intensity projection (MIP) images, and lower panels display the enlarged images of the nuclear region at the midplane. The yellow and cyan arrowheads point to the dividing nucleus and the cell boundary between the apical and basal cells, respectively. The corresponding stages are indicated, and the dotted line shows the stage when the nuclear division was completed, but the cell boundary was still unclear. (b and c) Graphs of the ratio of max/min of Feret's diameter (b) and circularity (c), which were measured using the mitochondria in the cell centre regions. From the time-lapse images, the time frames before, during, and after the zygotic division were used as mature, dividing and one-cell embryo, respectively. (d) Graph of the total mitochondrial area, measured using the whole-cell areas like in Figure 1h. The left illustration of b shows a schematic representation of the Feret's diameter, and the right illustrations show the correlation between the mitochondrial features and respective values. (e) 2PEM image of the time-lapse observation of the mitochondrial/nuclear marker in the presence of polymerization inhibitor for microtubules (1 $\mu$M oryzalin). Images are representative of five time-lapse observations. MIP images and enlarged images of the midplane are shown as similar to a. Orange arrowheads indicate the nucleus, which condensed but failed to completely divide. The letters on the graph indicate significant differences determined by the Tukey–Kramer test; $p < .05$ (b); $p < 0.01$ (c); not significant (d) ($n = 10$ for each stage). Scale bars: 10 and 1 $\mu$m (insets).

Figure 1b), and the mitochondrial shape values were similar in the meristemoid and SLGC (Figure 5p, q).

Overall, these findings showed that, not only the zygotic division, another type of asymmetric cell division is able to generate two daughter cells with unequal density of mitochondria. Nevertheless, the filamentous mitochondrial formation and its temporal separation during cell division are characteristic of the zygote. Therefore the zygote, which performs marked cell elongation and subsequent asymmetric cell division, might employ specific spatiotemporal regulation to unequally segregate the mitochondria.

## 4. Discussion

The combination of live-cell imaging with various dual-colour markers, pharmacological experiments, and extensive image quantification revealed the zygote-specific dynamic changes in mitochondrial morphology and distribution. The zygotic mitochondria formed long filaments along F-actin cables, which were temporally fragmented during zygotic division (Figure 4d). These shape changes were not detected in other tested tissues, suggesting that zygotic mitochondrial dynamics have specific mechanisms and roles.

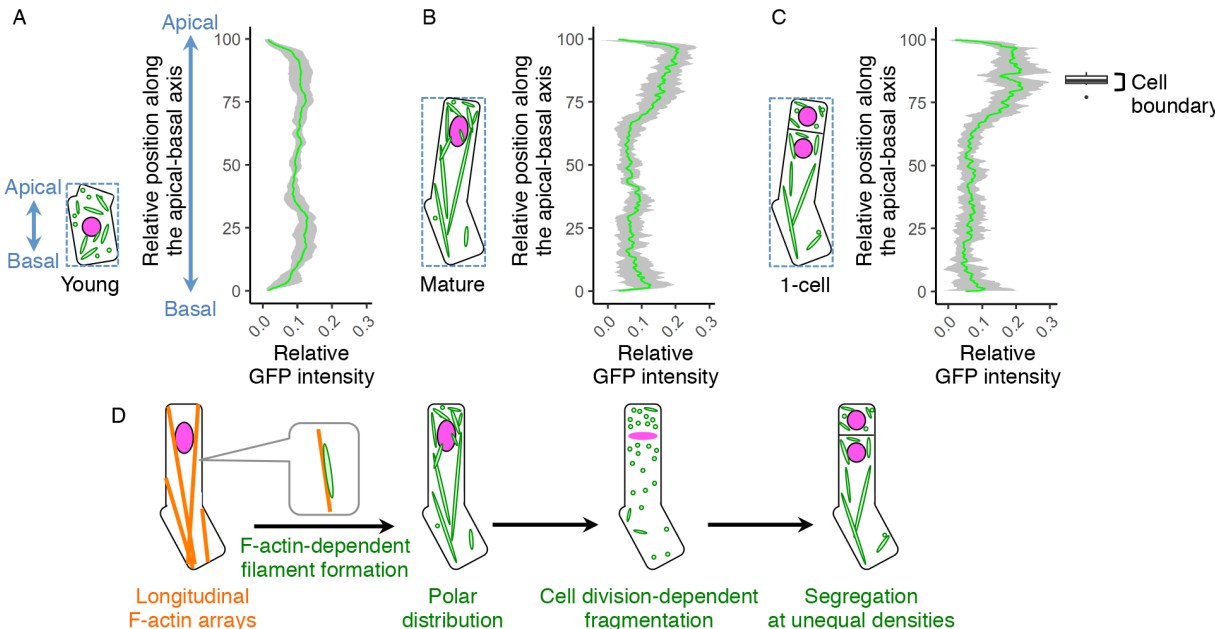

**Fig. 4.** Polar distribution and unequal inheritance of mitochondria along the apical–basal axis. (a–c) Graphs of the mitochondrial distribution along the apical-basal axis in the young zygotes (a), mature zygotes (b) and one-cell embryos (c). The images used for Figure 1g–i and the time-lapse images used for Figure 3b–d were used here to measure the mitochondrial green fluorescent protein (GFP) intensity. As shown in the left illustration of each graph, the apical–basal axis was determined using the bounding box (dotted square), and the apical cell tip and the basal bottom were set as the position 100 and 0, respectively. The GFP intensity shows the sum of the signals at each horizontal plane along the apical–basal axis. The right box plot of c shows the positions of cell boundaries of the apical and basal cells. (d) Schematic representation of the dynamics of the F-actin and mitochondria in zygote polarization and its asymmetric cell division. Grey bands represented as standard deviation (SD) [$n = 8$ (young) and 10 (mature and one-cell)].

### 4.1. The mechanisms underlying the dynamic shape changes in the zygotic mitochondria

We found mesh-like mitochondria in the egg cell and filamentous mitochondria in the zygote. The mitochondrial pattern in all cytoplasmic areas of the egg cell is consistent with the previous observations of both fixed and living materials (Gao et al., 2018; Wang et al., 2012; Yamaoka et al., 2011). On the other hand, the mesh-like pattern was found in living materials (Yamaoka et al., 2011), whereas the fixed egg cells were occupied by small dotted mitochondria (Gao et al., 2018; Wang et al., 2010). This discrepancy might be due to high fragility of this fine structure, similar to the thin tubular strands of zygotic vacuoles, which are apparent in living samples but not observed in fixed materials (Kimata et al., 2019; Mansfield & Briarty, 1991; Ueda et al., 2011).

Our pharmacological experiment revealed that the proper F-actin cables are necessary to form the filamentous structure of the zygotic mitochondria. The F-actin array itself, however, is not sufficient to form filamentous mitochondria, because that the F-actin cables align longitudinally in various cells, including all of our tested elongated cell types that harboured mitochondrial dots (Voigt et al., 2005). In principle, the filamentous mitochondria may be formed in several steps: the mitochondria associate with F-actin, they migrate along F-actin for assembly, and the fusion of adjacent mitochondria is promoted. The first two steps seem insufficient to induce filamentous mitochondria, because mitochondrial association and migration on F-actin, without forming filamentous structure, were reported in various cell types, including *Arabidopsis* root hair and tobacco culture cell (Doniwa et al., 2007; Zheng et al., 2009).

The above suggests that mitochondrial fusion might be the key step in mitochondrial elongation. Indeed, we found that the core fission regulator, DRP3A, does not associate with long mitochondria in the zygote, implying reduced fission activity and/or increased fusion activity. The fission/fusion balancing can also trigger mitochondrial fragmentation during zygotic division, because the cell-cycle dependent phosphorylation of DRP3A is known to promote mitotic fission of mitochondria in tobacco culture cells (Wang et al., 2012).

It would be also important to know the driving force of polar distribution of the mitochondria. We previously showed that the polar vacuole positioning at the basal cell region supports the nuclear localization at the apical cell tip (Kimata et al., 2019). Therefore, the vacuole might also support the mitochondrial distribution; that is, the large vacuoles displace mitochondria from the base cell end. Alternatively, they might independently and coordinately migrate towards the opposite cell ends, by forming the thin vacuolar strands and filamentous mitochondria to directly associate with F-actin.

Future studies are necessary to understand the molecular mechanisms underlying polar distribution and stage-specific mitochondrial changes in the zygote—for example, high-resolution imaging to monitor the exact timing of zygotic mitochondrial fusion/fission; simultaneous visualization of mitochondria, F-actin, vacuole, DRP3A, and its possible regulators, such as ELONGATED MITOCHONDRIA1, which interacts with DRP3A to support its accumulation to the mitochondrial fission sites (Arimura et al., 2008). A particularly important future study is the one that identifies the mitochondrial fusion regulators in plants to monitor their dynamics in the zygotes.

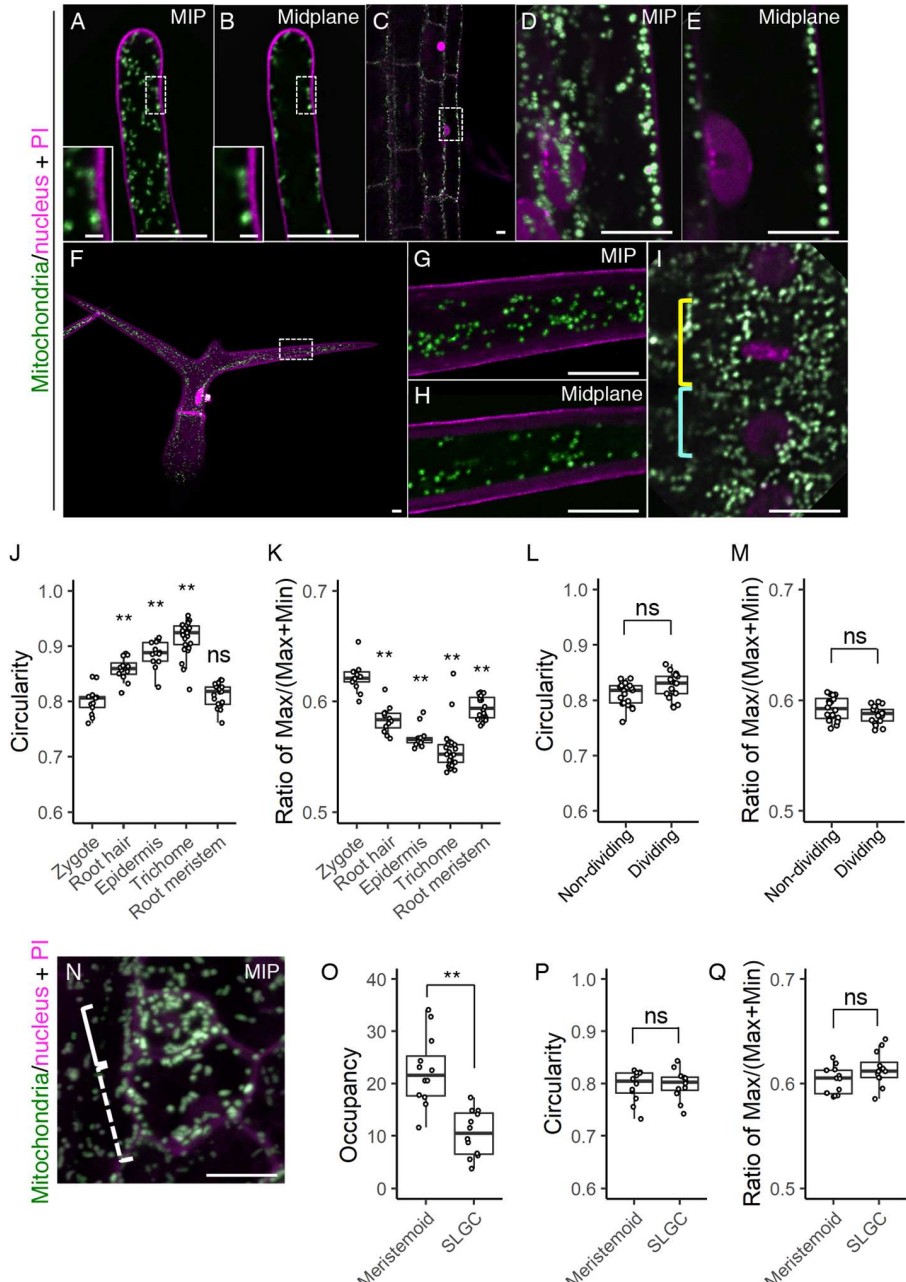

**Fig. 5.** Mitochondrial morphology and distribution in postembryonic organs. (a–i) Two-photon excitation microscopy (2PEM) images of propidium iodide (PI)-stained root hair (a and b), root epidermis (c–e), leaf trichome (f–h), and root meristematic cells (i) in 7-day-old (a–e, and i) or 10-day-old (f–h) seedlings expressing the mitochondrial/nuclear marker. Maximum intensity projection (MIP) (a, d, f, and g) and midplane (b, c, e, h, and i) images are shown. Insets in a and b show the enlarged images of the dotted square areas. The enlarged images of the dotted square areas in c and f are shown in d and e, and g and h, respectively. Yellow and cyan rectangles in i indicate the dividing and non-dividing cell, respectively. (j–m) Graphs of the circularity (j and l) and the ratio of max/(max+min) of Feret's diameter (k and m), which were measured using the mitochondria in a focal plane of the indicated cells. The cell types were indicated. The samples of the root meristem and the zygote in j and k correspond to those of non-dividing in l and m, and the zygote (DMSO) in Figure 2g, respectively. (n) 2PEM image of PI-stained stomatal precursor cells in 7-day-old seedlings expressing the mitochondrial/nuclear marker. MIP image was shown. Solid and dotted rectangles in N indicate the meristemoid and stomatal lineage ground cell (SLGC), respectively. (o–q) Graphs of the occupancy (o), circularity (p), and the ratio of max/(max+min) of Feret's diameter (q) of the mitochondria in the meristemoid and SLGC. Significant differences from the values of the zygotes, and non-dividing cells were determined by Dunnett's test (j and k); $^{**}p < .01$; ns, not significant [$n = 13$ (zygote), 12 (root hair), 13 (root epidermis), 25 (trichome) and 20 (root meristem)]. Significant differences were determined by Student's $t$ test (l, m, and o–q); $^{**}p < .01$; ns, not significant [$n = 20$ (non-dividing) and 17 (dividing) in l and m, and $n = 12$ in each cell type in o–q]. Scale bars: 10 and 1 $\mu$m (insets).

## 4.2. What are the roles of mitochondrial dynamics in the zygote?

We found that polar mitochondrial distribution in mature zygotes correlates with the unequal inheritance of the mitochondria in the daughter cells. The apical cells receive densely packed mitochondria, which would help these cells retain higher proliferation activities and initiate various developmental programmes (Gooh et al., 2015). It was supported by our finding that the stem cell-like stomatal meristemoid also contains densely packed mitochondria. This unequal inheritance after the zygotic division may be the result

of the combination of F-actin-dependent extension of filamentous mitochondria along the apical–basal axis and its temporal separation during cell division (Figure 4d). The suggestion that the mitochondrial distribution in the zygote pre-determines the inheritance pattern is supported by the *Arabidopsis miro1* mutant, the zygote of which contains abnormally enlarged mitochondria and the apical cell inherits a relatively fewer number of mitochondria (Yamaoka et al., 2011).

In addition to the unequal mitochondrial inheritance, mitochondrial dynamics in the zygote might contribute to various other aspects, because the stomatal precursor cells showed unequal mitochondrial density without forming long filamentous mitochondria. For example, the crucial roles of mitochondrial morphological change on bioenergetic reactivation were reported in *Arabidopsis* seed germination (Paszkiewicz et al., 2017). The dormant dry seed contains dotted and rudimentary mitochondria, but mitochondria are fused to form meshwork pattern after seed imbibition, resulting in bioenergetic and metabolic reactivation. Temporal mitochondrial fusion is also observed in the protoplasts just before cell division begins, and in cells under low levels of light, oxygen, or sucrose (Jaipargas et al., 2015; Ramonell et al., 2001; Sheahan et al., 2005). Therefore, the long fused mitochondria in the zygote likely help reactivate bioenergetics to initiate ontogeny.

In addition, the polar distribution of zygotic mitochondria may support cell polarization and axis formation, like in the *C. elegans* zygote, the mitochondria of which produce a local increase of hydrogen peroxide ($H_2O_2$), thus disrupting cell symmetry and forming the anterior–posterior axis (De Henau et al., 2020). In *Arabidopsis*, $H_2O_2$ and calcium are the key molecules leading to polar cell growth in the pollen tube, and recent work showed that mitochondrial calcium uniporter, MCU1/2, is required to support pollen tube elongation (Selles et al., 2018).

In spite of various possibilities of mitochondrial roles, our current technologies are still too poor to examine the direct contributions of the spatiotemporal behaviours of the mitochondria on each activity. For example, the actin polymerization inhibitor LatB effectively disrupted the mitochondrial shape in the zygote, but it destroys the F-actin array itself and thus broadly affects the F-actin-dependent dynamics, such as the polar positioning of nuclei and vacuoles in the zygotes (Kimata et al., 2016; Kimata et al., 2019). Therefore, we need to develop the new methods to disrupt the specific mitochondrial features and to monitor the consequent effects on various mitochondria-related activities, including bioenergetic reactivation and daughter cell fate after the asymmetric division of the zygote. It would be also helpful to determine the affected processes in *miro1* apical cell, which inherits fewer mitochondria and shows developmental arrest at this stage (Yamaoka et al., 2011).

## Acknowledgements

We thank Tomomi Yamada, Yumi Kuwabara and Terumi Nishii for their technical support; and Masaru Fujimoto, Shin-ichi Arimura, Nobuhiro Tsutsumi and Shohei Yamaoka for providing mitochondria markers. The microscopy was supported by the Live Imaging Centre at the Institute of Transformative Bio-Molecules (WPI-ITbM) of Nagoya University. We would like to thank Editage (www.editage.com) for English language editing.

**Financial Support.** This work was supported by the Japan Society for the Promotion of Science [Grant-in-Aid for JSPS Research Fellow (Y.K., grant number JP19J30006); Grant-in-Aid for Scientific Research on Innovative Areas (M.U., grant numbers JP17H05838, 19H04859, 19H05670, and 19H05676),

(T. Higashiyama, grant numbers JP16H06465, JP16H06464, JP16K21727), (T. Higaki, grant numbers JP18H05492, JP16H06280 (Advanced Bioimaging Support)); Grant-in-Aid for Scientific Research (B) (D.K., grant number JP17H03697), (M.U., grant number JP19H03243), (T. Higaki, grant number JP20H03289); and Grant-in-Aid for Challenging Exploratory Research (D.K., grant number JP18K19331), (M.U., grant number JP19K22421)] and the Japan Science and Technology Agency [PRESTO programme (D.K., grant number JPMJPR18K4)].

**Conflicts of Interest.** The authors declare no conflicts of interest.

**Authorship Contributions.** T. Higashiyama and M.U. designed the research; Y.K., N.A., H. M., and M.U. carried out the experiments; Y.K., T. Higaki, and D.K., analysed and visualized the data; and Y.K., T. Higaki, D.K., and M.U. wrote the manuscript.

**Data and Coding Availability Statement.** The data that support the findings of this study are available from the corresponding author, M.U., upon reasonable request.

**Supplementary Materials.** To view supplementary material for this article, please visit http://dx.doi.org/10.1017/qpb.2020.4.

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
