## [Reviewer Report]

June 30th, 2020

Olivier Hamant

Editor-in-Chief

Quantitative Plant Biology

Dear Editor:

I wish to submit an original research article for publication in Quantitative Plant Biology, titled “Mitochondrial dynamics and segregation during the asymmetric division of Arabidopsis zygotes.” The paper was coauthored by Yusuke Kimata, Takumi Higaki, Daisuke Kurihara, Naoe Ando, Hikari Matsumoto, and Tetsuya Higashiyama. 

This study reveals the dynamics of how the plant zygote segregates its mitochondria into the two daughter cells—the apical and basal cells—and shows that mitochondria are distributed unequally. Mitochondria in the zygote formed an extended structure associated with the longitudinal array of actin filaments (F-actins) and were polarly distributed along the apical-basal axis. We believe that our study makes a significant contribution to the literature because the mitochondria are the main organelles that provide the cell with energy, and therefore our finding that the mitochondria are differently inherited in the first two daughter cells has large implications for plant ontogeny.

Further, we believe that this paper will be of interest to the readership of your journal because we show various kinds of methods to quantify the real-time dynamics of the mitochondria in the zygote, and their unequal distribution during the cell division. This is a promising approach, and can be used in future studies utilising image quantification.

This manuscript has not been published or presented elsewhere in part or in entirety and is not under consideration by another journal. We have read and understood your journal’s policies, and we believe that neither the manuscript nor the study violates any of these. Details about competing interests are provided separately.

Thank you for your consideration. I look forward to hearing from you.

Sincerely,

Minako Ueda

Ph.D., Professor

Graduate School of Life Sciences, Tohoku University, Sendai, 980-8578, Japan

Tel: +81-22-795-6713

E-mail: minako.ueda.e7@tohoku.ac.jp

---

## [Reviewer Report]

*Comments to Author*: Making use of a pioneering set-up for live imaging of Arabidopsis embryos, the Ueda lab has made seminal contributions to our understanding of the cell biology underpinning the directed expansion and asymmetric division of plant zygotes. Previous work has investigated the role of actin filaments and microtubules in these processes and described the dynamics of the vacuolar compartment in detail. Here, Kimata and colleagues report on the dynamic organization of mitochondria. In elongating zygotes, mitochondria were found to form an actin-associated filamentous network that is fragmented prior to the first division; mitochondria were concentrated in the apical portion of mature zygotes, such that the apical daughter cell inherited a denser population. Several parameters, such as the total volume occupied by mitochondria, their average shape (circularity, Feret’s diameter), their association with actin filaments, and their spatial orientation with respect to the cellular axis, were quantified for the pertinent developmental stages. The work has been done with much attention to detail and is documented in excellent pictures/movies. Overall, I enjoyed reading the manuscript, finding it compelling and informative.

Specific comments/suggestions:

Line 33/34, “as new organelles cannot be synthesized”: This sounds awkward – new organelles are generated all the time (although it is true that plastids and mitochondria have to be inherited and cannot be synthesized de novo).

Line 34/35, “principal energy source organelles”: principle source of ATP biogenesis?

Line 40 and all over the manuscript, “more concentrated mitochondria”: The wording makes it sound like “concentration” is a property of the mitochondria. What seems to be happening is that the mitochondria are packed more densly in the apical part of the zygote (probably due to the presence of a central vacuole in the basal part of the zygote) and that the concentration of mitochondria (with respect to total cell volume, that is including the volume of the vacuole) is higher in the apical than the basal cell.

Line 53: This is a very strong statement – is the apical/basal axis really “defined” by the division of the zygote?

Line 63: Presumably the function of mitochondria is essential at all stages of the life cycle?

Line 66: Energy consumption does not necessarily mirror the rate of proliferation.

Line 96/70: The distribution of mitochondria with the first division has been documented before; for example, Manfield & Briarty (1991, cited in the manuscript) measured the distribution of plastids and mitochondria on the basis of electron micrographs and Yamaoka & al. (2011, also cited in the manuscript) measured the fluorescence associated with mitochondria in wild type and miro mutant zygotes and early embryos.

Line 77, “pre-existing mitochondrial pattern”: distribution pattern?

Line 79, “extended filamentous shape”: filamentous network?

Line 82, “these mitochondrial features”: similar features?

Lines 134 and 145: Not sure what “disorganized” is describing here – random distribution?

Lines 134/135 and 152/153: The title mentions “mitochondrial volume”, but later “cell region” and “total mitochondrial area” are used – this should be clarified.

Line 189/190: I am not familiar with the ZYGOTE-ARREST1 mutant and the Guo paper is not the references. Isn’t the APC/C a multi-protein complex? What does the gene encode?

Line 192, “contribute to the zygote”: contribute to zygote development?

Line 242-244: Similar to the nucleus displacing mitochondria in the center of elongating zygotes, it seems to me like this polar distribution arises because the large vacuole displaces mitochondria at the base of mature zygotes. This seems like a very simple mechanism for explaining the observed results (maybe the concentration of mitochondria is actually relatively constant with respect to the cytoplasm). Could the authors comment on this possibility either here or in the discussion?

Line 303, “are formed”: may be formed?

Line 338, “major energy source”: major source of ATP?

Line 354-363: Could miro mutants shed light on the question whether or not unequal distribution of mitochondria is important for further development? According to Yasmaoka & al., miro has no impact on polarized zygotic elongation or subsequent asymmetric zygotic division, but mutant apical cells have fewer mitochondria – do they still develop normally?

Figure 1, D,E,F: How were the threshold-values for “digitizing” determined? What is the influence of different threshold values on the measurements (how robust are the measurements)?

Figure 2, G: How was circularity calculated?

Figure 3, “Temporal fragmentation” – temporary fragmentation?

Figure 3, B and Figure 5, K: A ratio of “Max” to the sum of “Max” and “Min” would be a more straightforward measure (less prone to hyperbole).

Figure 3, C: mature is 0.7, vs. 0,8 in Figure 2 – what is the difference? Also, LatB treatment increases circularity to ~0.85, whereas in dividing zygotes it is <0.8 – so, less round?

Figure 3 E: From which area were the insets in the “MIP” row taken? What were the criteria for choosing these areas?

Figure 4: The Figurenicely summarizes the results!

---

## [Reviewer Report]

*Comments to Author*: In the current manuscript Ueda and co-workers have investigated the dynamics and morphology of mitochondria during the first division in the Arabidopsis zygote. The manuscript is well written and the experiments have been performed up to the highest standards. I appreciate their efforts to quantify the shape of the mitochondria in the zygote and how this changes during its asymmetric division. Since the authors have the biological material, have they performed time-lapse imaging looking at the zygote division in the lifeAct/mitoGFP double maker line? It would be nice to show what happens to the intensity of the actin marker compared to the intensity of the mitoGFP marker during the asymmetric division. Do the authors think that asymmetric divisions always could lead to an unequal division of the mitochondria or is this specific for the zygote. I appreciate that the authors have looked at the mitoGFP in different cells and tissues, but they did not provide any data of asymmetric divisions in other cell types such as stomata or in the XPP during lateral root initiation. It might be good to look at this to make the point whether the observed dynamics are specific for the zygote or rather linked to asymmetric divisions.

---

## [Reviewer Report]

*Comments to Author*: Dear Dr Ueda and colleagues, 

We have now received the comments from two expert reviewers on your manuscript. Please find their detailed comments attached. 

Both reviewers agreed on the interest and high quality of the work. 

They ask however for minor modifications prior acceptance to publication. 

As you will see, Reviewer 1 suggests a number of minor changes that could be addressed by rephrasing/modulating, in the results and the discussion text. Also, he asks to improve quantitative analysis of the data for several points. Notably improving shape quantification by normalizing better Feret’s diameters, which I think it is an important point to ensure no size effects for this shape quantifier. 

I think his different points could be addressed and/or discussed, and would improve the manuscript. 

Reviewer 2 is wondering if you have performed time-lapse imaging during the asymmetric division of the zygote for the life-act/mtGFP marker, to see their relative intensity. If you have the data, it could be nice to add such quantifications. 

He is also proposing to look at post-embryonic cell types presenting asymmetric cell divisions (ACD). Do stomata mother cells and/or lateral root founder cells display filamenteous/non spherical mitochondria before ACD as in the zygote? This would reinforce your conclusions on the specificity of mitochondria organisation in the zygote, or conversely, suggest a conserved mechanism for asymmetric divisions. This would broaden the biological significance of the results. The question should be possible to address in static imaging using the RPS5a::mtGFP marker reported here. 

However, please let us know if you see any technical issue to achieve this, that would over-delay resubmission of more than 2-3 months. 

In addition, I would like to thank the authors for the detailed Supplementary Materials file. However, in the imaging section, I’ve noticed that the wavelengths filters for emission detection used for the different fluorescent markers are missing, which could be important for the reader to reproduce experiments. 

We would be happy to receive a corrected version of your manuscript when it is ready. 

I thank you again for having submitted your excellent manuscript to Quantitative Plant Biology. 

Sincerely

Daphné

---

## [Reviewer Report]

Olivier Hamant

Editor-in-Chief

Daphné Autran

Associate Editor

Quantitative Plant Biology

September 3rd, 2020

Dear Olivier and Daphné,

We really appreciate for the positive comments and suggestions of the reviewers on our submitted manuscript. We think we have solved the issues inquired by reviewers. By adding new data and revising the phrases, we now have much accurate descriptions, as detailed in the responses to reviewers’ comments. We have also added the information for the wavelengths filters for emission detection used for the different fluorescent markers, which were pointed by Daphné. In addition, according to the instruction from Ms. Rebecca Fitchett, we have replaced the image for thumbnail, and integrated our Supplementary Materials and thus Supplementary References into the main text. The corrected parts in the revised manuscript are shown as track changes.

Thank you really so much for your great efforts. We are looking forward to hearing from you soon!

Yours sincerely, 

Minako Ueda 

Our original manuscript title was "Mitochondrial dynamics and segregation during the asymmetric division of Arabidopsis zygotes". 

Tracking #: QPB-20-0007

Authors: Kimata et al.

Minako Ueda

Ph.D., Professor

Graduate School of Life Sciences, Tohoku University, Sendai, 980-8578, Japan

Tel: +81-22-795-6713

E-mail: minako.ueda.e7@tohoku.ac.jp

---

## [Reviewer Report]

*Comments to Author*: The revised version comes with an expanded, more detailed methods section, and many small changes have been made throughout the text to improve clarity. This looks very nice, compliments to the authors.

---

## [Reviewer Report]

*Comments to Author*: Dear Minako, 

I would like to thank you very much for the detailed revision of your manuscript, addressing nicely all the comments. We recommend its publication in QPB. 

Best wishes

Daphné